# Catalyst-controlled stereodivergent synthesis of polysubstituted alkenes

Chengmi Huang [1,2,5], Dong Wu[1,5], Yu-Qing Zheng[3,5], Lujin Wang[1], Yuqiang Li [4], Yangyang Li [1,2] ✉, Wen-Bo Liu [3] ✉ & Guoyin Yin [1,2] ✉

The strategic utilization of earth-abundant transition metals in catalysis has emerged as a trans-formative movement in advancing of synthetic chemistry. Despite notable progress, the potential of leveraging diverse geometric configurations of these catalysts to achieve divergent synthesis remains largely untapped. In this work, we present a stereodivergent three-component borylfunctionalization of alkynes, enabled by ligand-modulated geometric variations in nickel catalysts. This approach provides a versa-tile platform for the stereodivergent synthesis of two classes of valuable polysubstituted alkene building blocks, a challenging feat in organic synthesis. Its practical utility is showcased by the rapid construction of biologically relevant molecules. Mechanistic studies support that different geometric configurations of nickel catalysts display distinct reactivity during key reaction steps, leading to the observed stereochemical outcomes.

The development of synthetic methods employing earth-abundant transition-metal catalysis has become a defining trend in the field of organic synthesis[1,2]. This progress is fueled by a growing commitment to sustainable chemical synthesis and the significant potential to discover and develop novel reactions that rare and precious metal catalysts cannot achieve[3,4]. In this regard, nickel catalysts have risen to prominence as a paragon, attributed to their diverse oxidation states— extending across the series from Ni (0) to Ni (IV) and their ambidextrous capacity to facilitate both 2e- and 1e- chemistry processes[5–8]. These features set the predominant logic for the reaction design in the realm of homogeneous nickel catalysis over the past several decades. In addition, it is well-known that organometallic complexes of the first-row transition metals exhibit diverse geometric configurations[9,10]. For instance, four-coordinate nickel (II) complexes typically exhibit either square planar or tetrahedral coordination geometries. In contrast, five-coordinate nickel (II) complexes tend to adopt either trigonal bipyr-amidal or square pyramidal geometries (Fig. 1a)[11,12]. These distinct coordination geometries significantly influence the reactivity and

selectivity of the complexes. Specifically, variations in the spatial structure and electronic effects of the ligands can lead to stereospatial differences when the substrate coordinates with the metal. This, in turn, plays a crucial role in the stereoselectivity of the reaction by modulating key elementary steps, such as oxidative addition and reductive elimination. However, the effective use of these properties for reaction development remains underexplored.

Polysubstituted alkenes are essential structural components found extensively in natural products, pharmaceuticals, and organic materials[13,14]. Different stereoisomeric forms of these alkenes often exhibit distinct activities, and in some cases, can have opposing effects in pharmaceutical applications (Fig. 1b). For example, Alitretinoin and Isotretinoin, two commercial drugs with opposite stereochemical configurations, display distinct therapeutic effects[15]. Similarly, Tamoxifen, used in breast cancer treatment for its antiestrogenic activity, has an opposite stereoisomer that exhibits estrogenic activity[16,17]. Therefore, precisely defining the geometric structure of these alkenes during synthesis is crucial for understanding their

[1]State Key Laboratory of Metabolism and Regulation in Complex Organisms, TaiKang Center for Life and Medical Sciences, The Institute for Advanced Studies, Wuhan University, Wuhan, Hubei, China. [2]Shenzhen Research Institute, Wuhan University, Shenzhen, China. [3]Hubei Research Center of Fundamental Science-Chemistry, Engineering Research Center of Organosilicon Compounds & Materials (Ministry of Education), Hubei Key Lab on Organic and Polymeric Opto-Electronic, Materials and College of Chemistry and Molecular Sciences, Wuhan University, Wuhan, Hubei, China. [4]Shanghai Artificial Intelligence Laboratory, Shanghai, China. [5]These authors contributed equally: Chengmi Huang, Dong Wu, Yu-Qing Zheng. ✉e-mail: yangyangl@whu.edu.cn; wenboliu@whu.edu.cn; yinguoyin@whu.edu.cn

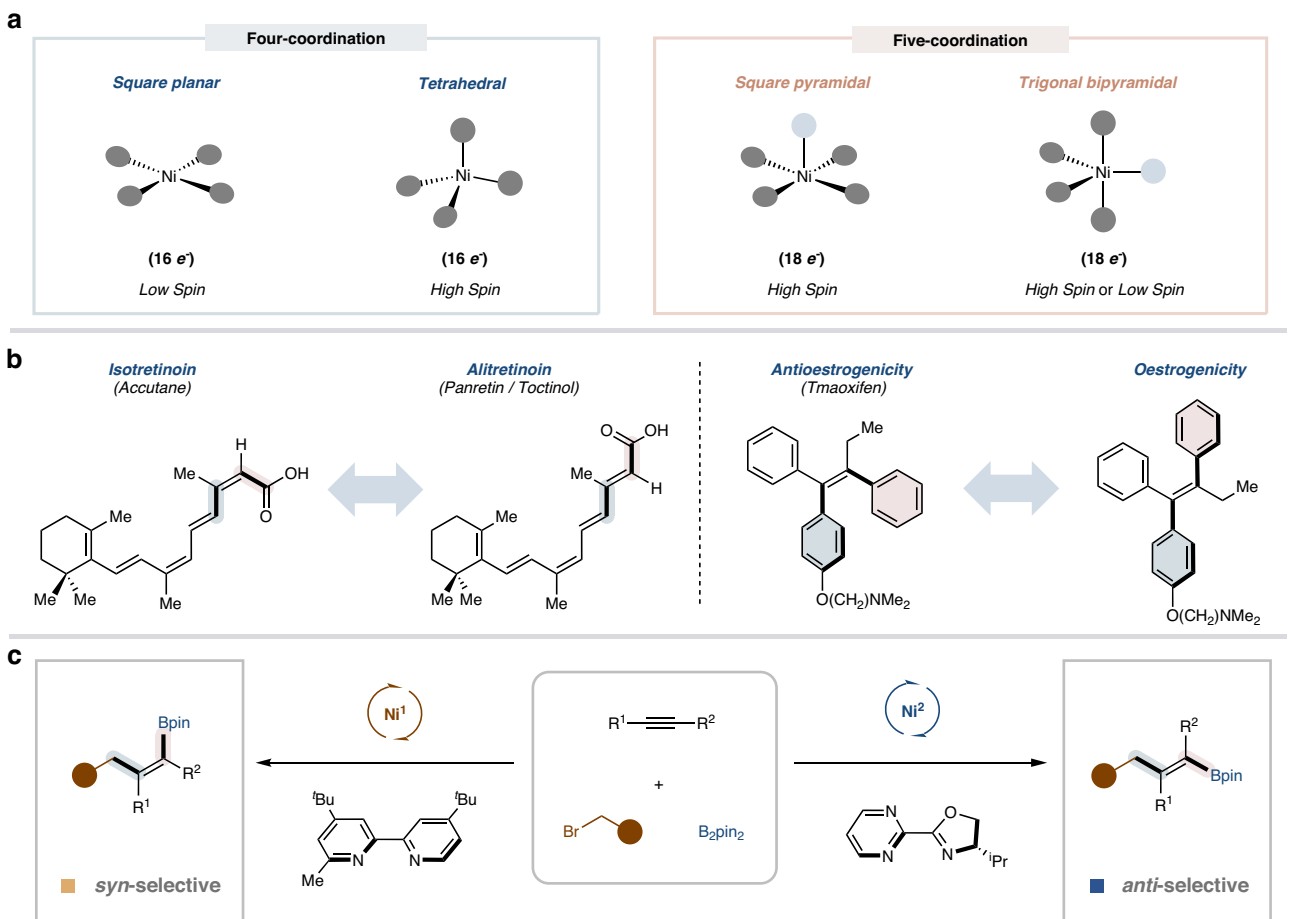

**Fig. 1 | Stereodivergent borylalkylation of alkynes. a** Ligand enabled geometric configuration differences of Ni (II) complexes. **b** Examples of how olefin geometry influences biological activity. **c** Catalyst-controlled stereodivergent borylalkylation of alkynes.

impact on biological activity in drug development campaigns. Consequently, it has stimulated significant interest in the precise synthesis of stereodefined polysubstituted olefins, which has led to substantial advancements in Wittig reactions[18], cross-coupling of alkenyl partners[19], alkene metathesis[20], functionalization of alkynes[21–25] or allenes[26–28], light-promoted alkene isomerization[29] and others. These methods typically require starting materials with well-defined stereochemistry or result in the production of only one stereoisomer. Stereodivergent synthesis of versatile polysubstituted alkenes has significantly transformed the landscape by providing a convenient and versatile platform for the efficient construction of complex molecules tailored for pharmaceutical targets[30–37]. However, the product space of existing strategies is still very limited. Therefore, developing effective strategies for the stereodivergent synthesis of versatile polysubstituted alkenes is still highly desirable.

Metal-catalyzed difunctionalization of alkynes represents an attractive and efficient platform for constructing polysubstituted alkenes using readily available starting materials[38–40]. Among these, carboboration stands out—the rich transformability of boron moieties[41,42] affords extensive scope for the synthesis of polysubstituted alkenes. However, alkyne carboboration reactions predominantly exhibit syn-stereoselectivity[43–45], thereby impeding the formation of *anti*-addition stereoisomers. While anti-selectivity can be obtained in a few cases through coordinating groups[46,47], steric hindrance[48], or alternative pathways involving radical intermediates[49], such approaches significantly restrict the reaction's substrate scope. In addition, stereodivergent synthesis that enables both syn and anti-selectivity remains underexplored. Building on our previous

investigation into metal-catalyzed functionalization of π-systems[50,51], particularly the successful development of Ni-catalyzed asymmetric *anti*-selective borylalkylation of terminal alkynes[52], we hypothesized that strategic ligand selection might effectively modulate the isomerization of alkenyl-nickel intermediates and their subsequent reactivity with electrophiles, potentially enabling stereodivergent synthesis. Herein, we demonstrate the validity of this hypothesis through the development of a nickel-catalyzed stereodivergent borylfunctionalization of alkynes, which highlights the crucial role of catalyst geometric configurations in controlling reaction stereoselectivity (Fig. 1c). In addition, this study introduces an efficient platform for the precise assembly of two classes of valuable tri- and tetrasubstituted alkenylboronates from simple feedstock chemicals by switching the supporting ligand. The observed ligand-dictated stereodivergence results from the distinct geometries of nickel catalysts, influencing elementary steps and consequently modifying stereochemical outcomes.

## Results
### Reaction development
Our study began by exploring the carboboration reaction of 4-ethoxyphenylacetylene (**1**), using pinacol diboronate (**2**) and benzyl bromide (**3**) as coupling partners. Extensive experimentations revealed that the choice of ligand is a crucial factor in determining the stereochemical outcome of this three-component reaction. As illustrated in Fig. 2, employing bioxazoline (**L1**), 1,2-diamine (**L2**), and terpyridine (**L3**) resulted in a mixture of *syn-* and *anti*-carboboration products (**4** and **5**) with low efficiency and poor stereoselectivity. The main

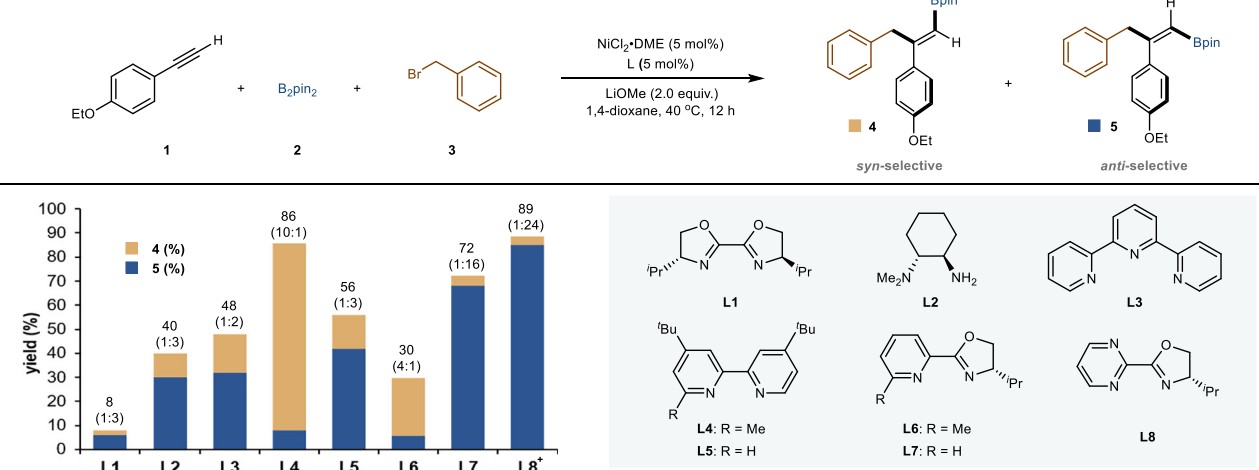

**Fig. 2 | Reaction discovery.** Identification of the optimal reaction conditions. NiCl₂·DME (5 mol %), **L** (5 mol %), **1** (0.4 mmol, 1.0 equiv.), **2** (0.8 mmol, 2 equiv.), **3** (0.6 mmol, 1.5 equiv.), and LiOMe (0.8 mmol, 2 equiv.) in 1,4-dioxane (2.0 mL), stirred at 40 °C for 12 h. GC yields, *syn:anti* selectivities were determined by GC analysis of the crude reaction mixture. ⁺**3** (0.8 mmol, 2.0 equiv).

byproducts were alkyne trimers and homocoupling of benzyl bromide. To our surprise, utilizing sterically hindered 2,2′-bipyridine (**L4**) and Pyox (**L6**) type ligand favoring the *syn*-selective product (**4**). Among them, **L4** yielded the best results with high stereoselectivity. In contrast, applying sterically unhindered 2,2′-bipyridine(**L5**), Pyox (**L7**), and *ⁱ*Pr-Pmrox type ligand (**L8**) resulted in a reversal of the stereoselectivity pattern, with **L8** yielding the optimal efficiency and selectivity for the *anti*-selective product (**5**). These results underscore the nature of the ligand in dictating the stereoselectivity of this nickel-catalyzed alkyne carboboration reaction.

## Substrate scope

With the optimized reaction conditions in hand, we explored the substrate scope of the *syn*-selective borylalkylation of alkynes, with the results summarized in Fig. 3. This catalytic system exhibited remarkable stereoselectivity and excellent functional group tolerance across a broad range of substrates. The reaction demonstrated good performance with various terminal aryl alkynes and aliphatic alkynes containing diverse functional groups (**6–18**). More significantly, the system proved to be versatile enough to address challenging cases, as evidenced by the successful conversion of both symmetric and unsymmetric internal alkynes into the corresponding tetrasubstituted alkenes[53] (**19–24**) with moderate to excellent stereoselectivity under identical reaction conditions. Further exploration revealed that varying the benzyl bromide (**25–30**) or switching it to α-bromophosphate, propargyl bromide, secondary α-bromoester, α-bromoamide, or allyl bromide also generated the corresponding syn-selective products (**31–35**) with high stereoselectivity. The stereochemistry of the *tri*- and tetrasubstituted alkenes was confirmed by X-ray diffraction analysis of representative products **24** and **28**.

We next extended our investigations to the substrate scope of the *anti*-selective carboboration reaction. As illustrated in Fig. 4, similar to the *syn*-selective carboboration reaction, aryl and heteroaryl alkynes with a variety of electronically diverse substituents (**36–44**) were efficiently converted to the corresponding trisubstituted alkenylboronates, achieving high yields and excellent *anti*-selectivities. A wide range of functional groups, such as ester (**37**), ketone (**38**), and sulfonyl (**39**), bromide (**40**) and boronate (**41**), were well-tolerated, offering significant versatility for subsequent cross-coupling reactions. Moreover, aliphatic alkynes tethering various substituents exhibited

excellent compatibility without compromising reactivity and selectivity (**45–51**). Moreover, the scope was not limited to terminal alkynes; internal alkynes were also compatible, delivering tetrasubstituted alkenes with good yields and *anti*-stereoselectivities (**52–54**). Further examination of benzyl halides bearing a range of functional groups (**55–61**) and hetero-aromatic benzyl bromides (**62–65**) revealed that all these organohalides underwent smooth carboboration reactions, producing the corresponding alkenes with moderate to excellent yields, as well as good regio- and stereoselectivity. Importantly, α-bromophosphate (**66**), propargyl bromide (**67**), along with α-bromoester (**68**), α-bromoamide (**69**), and allyl bromide (**70**), were all suitable coupling partners in this nickel-catalyzed system, resulting in the desired *anti*-selective products with good to excellent yields and stereoselectivities. In addition, the mass balance under moderate reaction yields was accounted for by alkyne trimerization (See Supplementary Information Section 5 for details). The structure and stereochemistry of the products were confirmed by X-ray diffraction analysis of compounds **52** and **58**.

## Synthetic applications

To demonstrate the synthetic utility of this stereodivergent chemistry, we applied it to the synthesis of biologically relevant compounds (Fig. 5). First, two distinct stereoisomers, **6** and **36**, obtained from this reaction, which were subsequently subjected to Matteson homologation and oxidation[54]. This transformation efficiently produced both (*E*)- and (*Z*)-allyl alcohols **71** and **72** (Fig. 5a), representing a significant improvement over previously reported methods that required 4 and 6 synthetic steps for their respective preparations. These alcohols serve as key intermediates in the synthesis of an agent with anticancer activity[55] and a human Icmt inhibitor[56], respectively. Second, we successfully synthesized key intermediate 74, which is crucial for the construction of the natural product Conulothiazole A[57], in four steps (Fig. 5b). This represents a significant improvement over the previously reported five-step sequence, which not only required additional synthetic steps but also involved the use of alkylcopper reagent and toxic OsO₄. More importantly, we achieved an even more streamlined synthesis of the natural product (*E*,*E*)-α-homofarnesene (**78**), completing its preparation in just two steps starting from commercially available materials (Fig. 5c). This remarkable reduction in step count, from the previously required five steps[58,59] to only two, highlights the efficiency and practicality of our methodology.

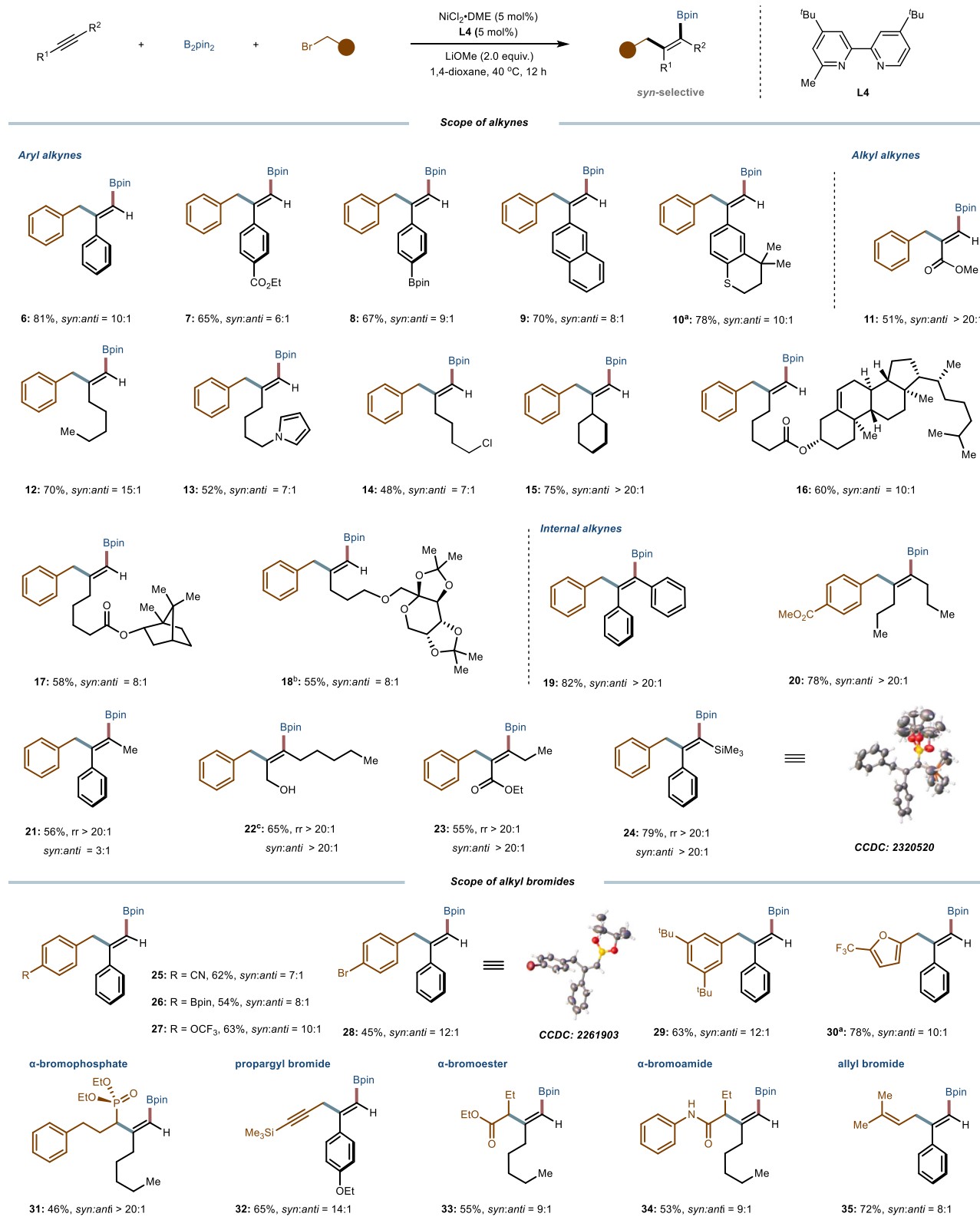

**Fig. 3 | Scope of the *syn*-selective carboboration reaction.** General reaction conditions: alkyne (0.4 mmol, 1.0 equiv.), B₂pin₂ (2.0 equiv.), alkyl bromide (1.5 equiv.), NiCl₂·DME (5 mol%), **L4** (5 mol%), LiOMe (2.0 equiv.) in 1,4-dioxane (0.2 M) was at 40 °C stirred for 12 h. The given isolated yields refer to the single isomeric product; *syn:anti* selectivities were determined by GC analysis of the crude reaction mixtures. [a] **L9** (6-methyl-2,2'-bipyrdine) was used. [b] CPME was used. [c] **L5** was used.

## Mechanistic study

To gain insights into the ligand-exerted dichotomy in this nickel-catalyzed three-component reaction, we carried out a series of mechanistic experiments. We found no evidence of isomerization for product **4** under the *anti*-selective reaction conditions, nor for product **5** under *syn*-selective reaction conditions (Fig. 6a). These results confirm that the observed *syn/anti*-stereoselectivity is not attributable to product isomerization. Furthermore, when phenylacetylene

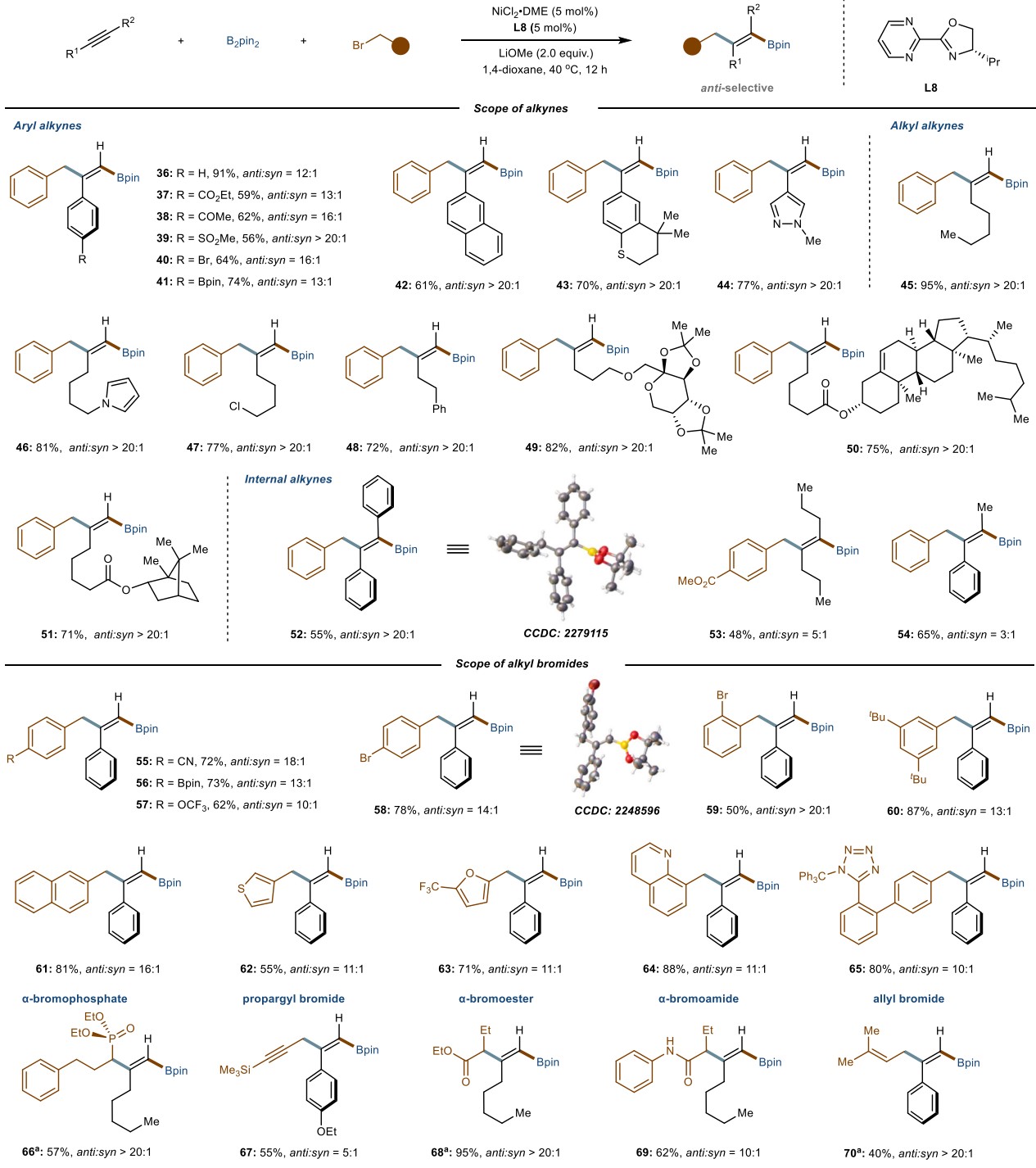

**Fig. 4 | Scope of the *anti*-selective carboboration reaction.** General reaction conditions: alkyne (0.4 mmol, 1.0 equiv.), B₂pin₂ (2.0 equiv.), alkyl bromide (2.0 equiv.), NiCl₂·DME (5 mol%), **L8** (5 mol%), LiOMe (2.0 equiv.) in 1,4-dioxane (0.2 M) was at 40 °C stirred for 12 h. The given isolated yields refer to the single isomeric product; *anti:syn* selectivities were determined by GC analysis of the crude reaction mixtures. [a] **L5** was used.

(**79**) and B₂pin₂ were reacted with H₂O instead of benzyl bromide, product **80** was obtained with a 7:1 *syn/anti* ratio under the *syn*-selective reaction conditions. Similarly, under *anti*-selective conditions, the reaction yielded product **81** with a 1.3:1 *anti/syn* ratio (Fig. 6b). A series of stoichiometric experiments were conducted using Ni(cod)₂ and two optimal ligands (**L4** and **L8**) with bromoalkenyl boronate (**82**) of different *Z/E* ratios as the starting material. When these reactions were quenched with H₂O, the resulting

product (**80**) was obtained with a low *syn/anti* ratio (Fig. 6c). Moreover, when benzyl bromide (**3**) was used instead of H₂O in these stoichiometric experiments, high *syn*- and *anti*-selectivity was achieved under the same conditions, regardless of the initial *syn/anti* ratio of **82** (Fig. 6d). These results strongly suggest the involvement of a rapid and reversible isomerization process in the catalytic cycle, which likely occurs prior to the oxidative addition step, and are consistent with the Curtin–Hammett principle[60,61], the

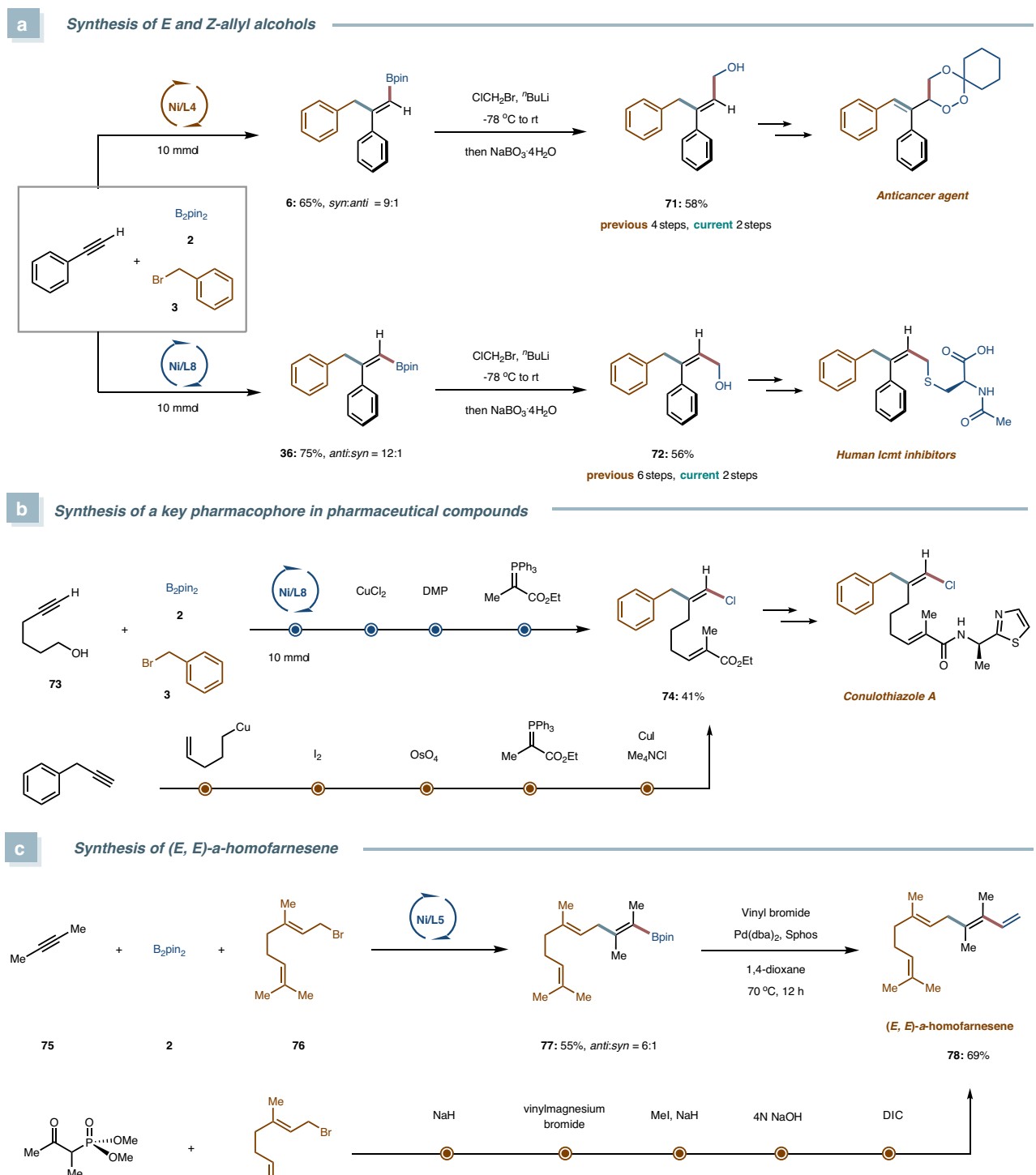

**Fig. 5 | Synthetic applications. a** Synthesis of *E* and *Z*-allyl alcohols. **b** Synthesis of a key pharmacophore in pharmaceutical compounds. **c** Synthesis of (*E, E*)-α-homofarnesene.

stereoselectivity is mainly determined by the oxidative addition and reductive elimination steps.

In addition, radical trapping experiment revealed that addition of TEMPO resulted in significant inhibition of the reaction, and the benzyl radical was identified as the TEMPO-trapped adduct (Fig. 6e). Radical clock experiment was performed with the alkylbromide derivative, and the ring-opened product was observed (Fig. 6f). These results indicated the involvement of an alkyl radical in the reaction system. Moreover, we conducted a series of electrochemical investigations

using cyclic voltammetry (CV)[62]. The CV measurement revealed a significant observation: the electrochemical profile obtained after the addition of B₂pin₂ and LiOMe closely matched that of a Ni(II) salt (Fig. 6g). This finding provides evidence against the possibility of Ni(II) reduction by the that the combination of B₂pin₂/LiOMe system, and suggests that the catalytic cycle is initiated through the formation of a Ni(II)-Bpin species.

Based on the above results, we propose that this multicomponent reaction is initiated by the migratory insertion of an alkyne into the Ni

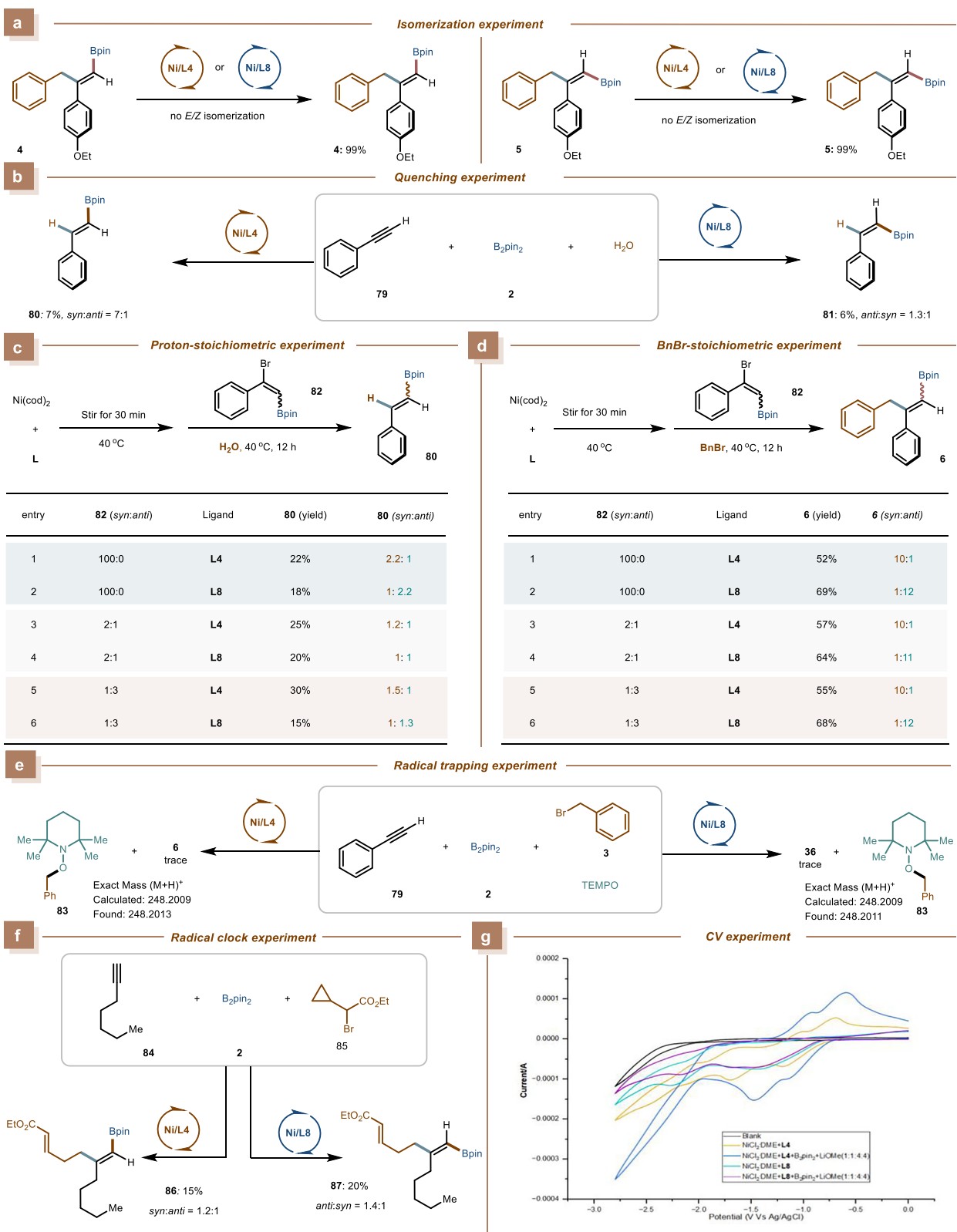

**Fig. 6 | Mechanistic study. a** Isomerization experiment. **b** Quenching experiment. **c** Proton-stoichiometric experiment. **d** BnBr-stoichiometric experiment. **e** Radical trapping experiment. **f** Radical clock experiment. **g** CV experiment.

(II)-Bpin species, generating a *syn*-alkenylnickel (II) intermediate, which then undergoes reversible isomerization to form an *anti*-alkenylnickel (II) species (Fig. 7)[63]. The ligand plays a crucial role in regulating the relative reactivity of these two nickel species toward alkyl radicals, thereby enabling the selective formation of the corresponding carboboration products and Ni(I) species. In addition, the Ni(I) species reacts with the alkyl halides **3**, producing the alkyl radical and regenerating the Ni(II) catalyst.

**Fig. 7 | Proposed mechanism.** The possible mechanism for catalyst-controlled stereodivergent synthesis of polysubstituted alkenes.

## DFT calculations

To further validate these findings, we investigated the isomerization process using density functional theory (DFT) calculations (Fig. 8). Three potential pathways were considered: (i) a zwitterionic carbene-type intermediate[64–66],(ii) an alkene-radical intermediate formed through reversible Ni-C homolysis[67], and (iii) a three-membered η²-vinylnickel transition state[68–72]. The first two pathways seemed unlikely (See Supplementary information Supplementary Figs. S19 and S20 for details). Instead, the DFT calculations suggest that isomerization most likely proceeds via the η²-vinylnickel transition state (Fig. 8a). Although it has relatively high energy, the radical chain initiation step surmounts a comparable energy barrier (Fig. 8b)[73–76], and the free radicals need to cross solvent cages[77], leading to a low concentration of alkyl radicals. As a result, the catalytic system is likely to reach equilibrium before the oxidative addition step, indicating that the stereoselectivity-determining step is governed by subsequent transformations, consistent with our mechanistic studies (Fig. 6b–d).

To further validate the origin of reaction stereoselectivity, additional DFT calculations were conducted. After comprehensively exploring the reaction potential energy surface and reaction pathways as thoroughly as possible, we found that the stable configuration of the divalent alkenylnickel species is significantly influenced by the ligand's nature. In the catalytic system with **L4** (Fig. 8c), using the ligand with a 6-substituent allows the divalent alkenylnickel complexes ᵀCP3 and ᵀCP4 to adopt either tetra-hedral or square pyramidal coordination geometries with a triplet (high-spin) state to alleviate steric hindrance caused by ortho-substituents[78,79]. Both configurations create a crowded environment for the radical addition step, making it quite sensitive to the alkenylnickel species' stereochemistry. **CP4**, with distorted square pyramidal geometry, is more prone to be attacked by free radicals than **CP3**, with tetrahedral geometry, as evidenced by a 1.6 kcal/mol energy gap between **TS3** and **TS4**. Through distortion/inter-action analysis, we found this difference comes from the distinct steric hindrance distributions of the two key intermediates, which was further corroborated by the surface distance projection map. (See Supplementary Information Supplementary Fig. S14 for details). This result indicates that the radical addition step is the stereochemistry-determining step for the *syn*-selective reaction. In contrast, in the reaction system with **L8**, the alkenylnickel

complexes ˢCP7 and ˢCP8 preferentially adopt a square-planar coordination geometry with a singlet (low spin) state (Fig. 8d). Unlike the tetrahedral and distorted square pyramidal coordination geometries, the square-planar coordination geometry results in a less congested environment. As a result, the radical addition step is minimally influenced by the stereochemistry of the alkenylnickel species, as evidenced by the negligible energy difference between the transition states **TS7** and **TS8**. Moreover, a 4.5 kcal/mol energy gap between **TS9** and **TS10** in the reductive elimination step indicates that this step is the key determinant of stereochemistry in the *anti*-selective reaction. We also investigated the reactivity of Ni(II) complex with different spin states and configurations. The alternative pathways are less favored due to the higher energy barrier.

## Discussion

In summary, we have successfully developed a nickel-catalyzed stereodivergent borylcarbofunctionalization of alkynes, high-lighting the potential of catalyst geometric configuration in modulating reaction stereoselectivity. This innovative strategy provides a convenient and efficient approach for synthesizing valuable tri- and tetrasubstituted alkenylboronic esters using readily available alkynes, alkyl bromides, and a diboron reagent, with exceptional levels of chemo-, regio-, and stereoselectivities. The method showcases remarkable functional group compatibility and a broad substrate scope, encompassing both terminal and internal alkynes as well as diverse alkyl bromides. The synthetic utility of this chemistry is underscored by its ability to streamline the synthesis of drug-relevant molecules.

## Methods

### Representative procedure for stereodivergent borylfunctionalization of alkynes

To an oven-dried 10 ml reaction tube equipped with a magnetic stir bar, the following were added in a nitrogen-filled environment: NiCl₂·DME (4.4 mg, 0.02 mmol, 5 mol%), **L4** or **L8** (0.02 mmol, 5 mol%), LiOMe (30.2 mg, 0.8 mmol, 2.0 equiv.) and B₂pin₂ (0.8 mmol, 2.0 equiv.). Then anhydrous 1,4-dioxane (1.0 mL), alkynes (0.4 mmol, 1.0 equiv.), alkyl bromides (0.6 mmol, 1.5 equiv.) and anhydrous 1,4-dioxane (1.0 mL) were added in this order, and the mixture was

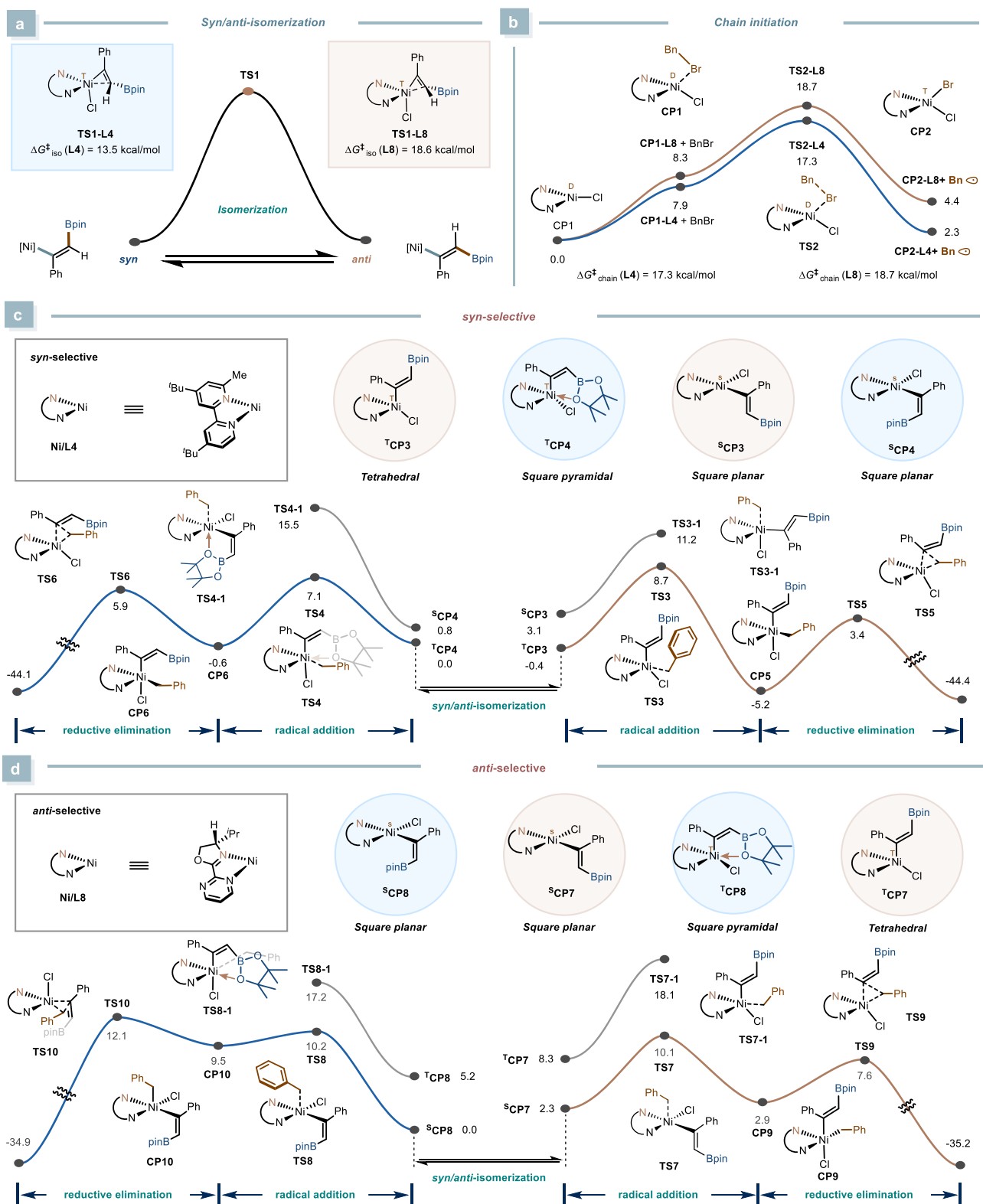

**Fig. 8 | DFT calculations. a** Computational analysis of *syn/anti* isomerization. **b** Computational analysis of chain initiation. **c** Computational analysis of *syn*-selective. **d** Computational analysis of *anti*-selective. DFT calculations were performed at the M06/def2-TZVP/SMD(1,4-dioxane)//B3LYP-D3(BJ)/def2-SVP of theory. *CP* complex; *TS* transition state.

stirred at 40 °C for 12 h. Upon completion, the reaction mixture was cooled to ambient temperature and subjected directly to preparative thin-layer chromatography to afford the corresponding product. For further details, please see the Supplementary Information.

## Data availability

All information relating to optimization studies, experimental procedures, mechanistic studies, DFT calculations, NMR spectra, and high-resolution mass spectrometry are available in Supplementary Information. All other data are available from the corresponding authors

upon request. Crystallographic data for the structures reported in this Article have been deposited at the Cambridge Crystallographic Data Center, under deposition numbers CCDC2320520 (**24**), 2261903(**28**), 2279115(**52**), 2248596 (**58**). Copies of the data can be obtained free of charge via https://www.ccdc.cam.ac.uk/structures/. Source data are provided in this paper.

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

## Acknowledgements

This work was supported by the National Natural Science Foundation of China (22122107 to G.Y., 22401220 to Yangyang Li, 22371215 and 22222111 to W.L.), the National Key R&D Program of China (NO.2022ZD0160100 to Yuqiang Li, 2022YFA1502902 to W.L.), the Fundamental Research Funds for the Central Universities (413100070 to Yangyang Li), Guangdong Basic and Applied Basic Re-search Foundation (2024A1515011689 to G.Y.), Scientific Research Innovation Capability Support Project for Young Faculty (ZYGXQNJSKYCXNLZCXM-H17 to G.Y.) and the Large Scale Instrument and Equipment Sharing Foundation of Wuhan University. We thank the Core Facility of Wuhan University for help with X-ray crystallographic analysis. The numerical calculations in this research have been done on the supercomputing system in the Supercomputing Center of Wuhan University. Professor Xiao-Tian Qi (WHU) is thanked for his supervision regarding the DFT calculations.

## Author contributions

G.Y. designed the project and directed the work; C.H. developed the catalytic method; C.H., D.W. and L.W. performed all synthetic experiments and Y.Z., Yuqiang Li, and W.L. performed all DFT calculations. G.Y., C.H., D.W. and Yangyang Li. wrote the manuscript.

## Competing interests

The authors declare no competing interests.
