## [Transparent Peer Review file · Nature Communications]

Catalyst-controlled stereodivergent synthesis of polysubstituted alkenes

Corresponding Author: Professor Guoyin Yin

Version 0:

Reviewer comments:

Reviewer #1

(Remarks to the Author)
comments enclosed as pdf

Reviewer #2

(Remarks to the Author)

The submitted article, entitled "Catalyst geometric variations for stereodivergent synthesis of polysubstituted alkenes," demonstrates the synthesis of syn and anti-products using Ni catalysts containing various ligand types, efficiently with L4 for syn and L8 for anti-products. Suginome et al. reported nickel-catalyzed cyclizative trans-carboboration of alkynes (Asian J. Org. Chem. 2013, 2, 968 – 976). More recently, Huang et al. reported regioselective trans- carboboration of internal alkynes via a nickel catalysis system with the aid of the directing group strategy (Chem. Sci., 2024, 15, 2236). The regioselectivity was accurately switched by the nitrogen ligand (terpy) and phosphine ligand (Xantphos). Based on these literatures it appears that the use of Ni catalyst for the reported reaction is not an unusual. Author must include these references. Yin et al. (same author) reported asymmetric anti-selective borylalkylation of terminal alkynes by nickel catalysis with wide substrate scope (Ref. 40).

In this report, authors used commercially available different nitrogen ligands to switch the regioselectivity. In term of ligand design this work lack originality. In my opinion, this report is showing some new aspects of Ni-catalyzed regioselective carboboration of alkynes but it lacks novelty and is not suitable for the publication in Nature Communications.

Specific comments:

1. In introduction section, authors should include relevant references related to organoboron compounds and polysubstituted vinyl boronates.
2. References related to carboboration reactions must be included, particularly related to Ni-catalyzed.
3. There are no phosphorous based ligands explored towards this alkyne carboboration. Any reason why nitrogen ligand over phosphorous.
4. Most of the carboboration product show moderate yields, is there any byproducts arising from the competent 2 component system or is it just starting material, if so higher temperature or time or catalyst loading could help?
5. Substrate scope of electrophile is very limited (only activated benzyl bromide), authors should explore unactivated alkyl halide for carboboration reaction. MeI should explore?
6. Authors should provide NMR evidence for Ni(II)-Bpin species, after the addition of B₂pin₂ and LiOMe to the Ni catalyst.
7. Cyclic voltammogram that author presented is questionable. They claimed that after the addition of B₂pin₂ and LiOMe to Ni catalyst, it is closely matched with the reported literature, but with the ligand it must have some change that can be seen with some difference. Authors can comment on this? Difference is more when B₂pin₂ and LiOMe were used in excess (4 equiv.). It is clear in CV, both peaks around -1.1 and -1.7 V are due to ligand effect. It needs to justify? Additionally, if CV data collected on 200 mV sec⁻¹ scan rate, then also current reaches up to 0.2 μA?
8. Some of experimental data should be recheck, ¹³C NMR of 4, spectrum frequency is not according to 600 MHz instrument? Are there compound 26 data collected on 400 MHz? For compound 29, if it is on 400 MHz, then 11B cannot be 193 MHz? Similarly, data need to be checked for more clarity.
9. HRMS data missing for a few compounds like for 6, 19, 24.
10. In DFT, author work on optimized gas phase structures along with solvent effect. Can author comment on this, why the

gas phase condition considers even all reactions performed at 40°C in 1,4-dioxane? Without solvent, what changes appear in the product as per DFT?

11. The DFT provides better insights on how the geometry influences the selectivity of the target product. However, the work lacks in providing better insights on the mechanism via experimental route. (refer <https://doi.org/10.1016/j.chempr.2022.10.003>) Also, no explanation or insights are provided on why the optimized ligand provides selectively one single isomer.

Reviewer #3

(Remarks to the Author)
review attached

Reviewer #4

(Remarks to the Author)

Yin and co-workers reported an interesting ligand-controlled stereodivergent borylalkylation of alkynes. The syn- and anti-carboboration products were obtained by using the sterically hindered 2,2'-bipyridine ligand (L4) and the iPr-Pmrox type ligand (L8), respectively. Through experimental and computational analysis, they claimed that the divergent stereoselectivities are attributed to the different geometric configurations of Ni catalysts. However, their presented computational results do not support their conclusions and the explanations remain vague. Main concerns are listed below.

1. Their explanation for the stereoselectivity is based on the Curtin-Hammett principle. However, the activation barrier of syn/anti isomerization is much higher than that of the subsequent steps, which indicates the reaction does not fulfill the Curtin-Hammett principle.
2. The geometric configurations of Ni catalysts were considered to be crucial for the reaction. Have the authors systematically analyzed all possible geometric configurations of Ni complexes, including the intermediates and transition states? For instance, interchanging the positions of the Cl, Bn, and alkenyl groups in CP5 will lead to different geometric configurations.
3. Related to the previous question, the reviewer noticed that CP10 adopted a different geometric configuration compared to CP9. CP10 has to undergo configuration isomerization for the reductive elimination step; however, CP9 can directly undergo reductive elimination. Why did the authors use different configurations here? Will the configuration affect the activation barrier? The authors should carefully consider all possible configurations, otherwise the comparison is biased and the conclusion that reductive elimination step decides the stereoselectivity in the anti-selective reaction is not reliable.
4. For the same reason, the distortion-interaction analysis (Figure S13) of the reductive elimination is not convincing. It's unsurprising that TS10 experiences larger distortion than TS9, since TS10 undergoes configuration isomerization compared to CP10 as I pointed out above.
5. Why do the two ligands, L4 and L8 lead to different preferred geometric configurations and preferred spin states for the Ni catalysts?
6. The explanation for the 1.6 kcal/mol difference between TS3 and TS4 stays vague. The statement "the distorted trigonal bipyramidal geometry, which exhibits greater stretching than the tetrahedral geometry, is more prone to be attacked by free radicals" needs more support, such as structural analysis.
7. The ligand screening experiments in Figure 2 demonstrate that the substituents also play an important role in controlling the stereoselectivity, for instance, reversed stereoselectivities were obtained by using L4 and L5 lead and using L6 and L7, respectively. Can these substituent effects be explained by their mechanistic model?

Other comments:

1. The legend of Fig.1c (Catalyst geometric variations for stereodivergent borylalkylation of alkynes) seems inappropriate, as it's hard to see the difference in catalyst geometries for the two systems.
2. Page 4, line 82, it should be "ligand (L4)" not "ligands (L4)"
3. L5 and L6 are not mentioned in the main text.
4. The geometric configurations in Fig 7 are unclear and difficult to differentiate, especially for TCP4 and TCP8. The authors label TCP4 as trigonal bipyramidal and TCP8 as square pyramidal, but they appear identical in drawing. The authors should carefully revise the structures in Fig 7 to ensure that the ligands in the apical/axial and equatorial positions are clearly distinguishable. Moreover, based on the coordinates in the SI, the geometric configuration of TCP4 is closer to square pyramidal. The deviation of Cl from the equatorial plane appears to be due to steric hindrance at the Bpin position.
5. What is the spin state for TCP8? The label for spin state is missing in TCP8.
6. The writing may need further improvement. Some sentences contain grammatical errors or are unclear. For example, the sentence "Although it relatively high energy, the radical chain initiation step surmounts a comparable energy barrier (Fig. 7b), and the free radicals need to cross solvent cages, leading to a low concentration of alkyl radicals." contains a grammatical error and should be revised for clarity. Additionally, in the sentence "We also discovered the reactivity of Ni(II) complex with different configurations which are raised by different spin states," the word "discovered" may be better replaced with "studied" or "investigated".

Version 1:

Reviewer comments:

Reviewer #1

(Remarks to the Author)

Authors addressed the concerns; consequently, I recommend the manuscript for publication in its current state.

Reviewer #2

(Remarks to the Author)

The submitted revised article, entitled "Catalyst geometric variations for stereodivergent synthesis of polysubstituted alkenes," demonstrates the synthesis of syn and anti-products using Ni catalysts containing various ligand types, efficiently with L4 for syn and L8 for anti-products. The authors have addressed most of the technical concerns raised by the reviewers; however, my primary concern pertains to the novelty of the work.

Reviewer #4

(Remarks to the Author)

The authors have addressed my questions. I support its publication on Nature Communications.

Response to Reviewers Comments

Reviewer 1:

The paper discusses the nickel-catalyzed stereodivergent carboborylation of alkynes, demonstrating the possible role of the catalyst's geometry in modulating reaction stereoselectivity. The synthetic utility of this chemistry is demonstrated by the synthesis of drug-relevant molecules. A detailed mechanistic study including DFT calculations was carried out to support the proposed mechanism. Overall, the ligand-controlled stereodivergent synthesis is impressive.

Recommendation: The syn-selective carbofunctionalization including the carboborylation of alkynes, is well known, however, the anti-selective carboborylation is uncommon. The same author reported the anti-selective carboborylation of alkynes (10.1021/jacs.3c05969). The difference is a change in the electrophile; i.e., α -bromo carbonyls vs. benzyl bromides. Since the current work combines strategies and demonstrates the ligand-controlled stereodiversity, also provides a detailed mechanistic study, I recommend it for publication after a major review.

Our response: We do appreciate the reviewer for his/her positive comments and support for publishing our work in *Nature Communications*.

(1) In reaction development, the author should disclose the byproducts observed during the optimization

Our Response: We thank the reviewer for this valuable suggestion. The main byproducts observed during reaction optimization were alkyne trimers and homocoupling of benzyl bromide. A detailed description has been added to the *Reaction Development section*.

(2) In Fig. 3 and Fig. 4, the reason for lower yields for substrates like 70 was not discussed. Did the author observe byproducts?

Our Response: We thank the reviewer for this professional suggestion. In response, we carefully analyzed the reactions that afforded low yields. As shown below, alkyne trimers were identified as the major side products in two representative cases. Additionally, we observed protodehalogenation and electrophile-derived homocoupling byproducts. These findings have been incorporated into the main text, and the corresponding experimental results have been included in the revised Supplementary Information (Section 5, page S26).

Wuhan University

武 汉 大 学

(3) Apart from activated halides, did the author attempt a reaction with unactivated alkyl halides (bromides or iodides)?

Our Response: We thank the reviewer for this valuable suggestion. In response, phenylpropyl halides and methyl iodide were tested under the standard conditions. However, the desired three-component coupling product was not obtained. Instead, the major products were alkyne trimers and dehalogenated byproducts, likely due to the low reactivity of these alkyl electrophiles.

(4) Page number 10: line 172-174, the author stated that “These results strongly suggest the involvement of a rapid and reversible isomerization process in the catalytic cycle, which likely occurs prior to the oxidative addition step, and are consistent with the Curtin–Hammett principle”. This is a bit confusing for me because the isomerization occurs from the vinylnickel intermediate; even if it reacts with water or benzylbromide, the stereochemical outcome should be the same because the ligand is what dictates the stereochemistry (See Fig.6c and 6d).

Our Response: We thank the reviewer for this insightful comment. Both experimental studies and DFT calculations suggest that the isomerization of alkenylnickel intermediates is rapid and reversible and the stereoselectivity of the reaction is primarily determined by the radical addition (Ni/L4) and reductive elimination (Ni/L8) steps, respectively. Unlike the benzyl group, a proton—due to its minimal steric bulk—cannot effectively differentiate between the two alkenylnickel

Wuhan University

武 汉 大 学

intermediates. We apologize for any confusion this may have caused and have revised the relevant section accordingly. We hope this clarification adequately addresses the reviewer's concern.

(5) Page number 10: line 178, the author states that “This finding provides evidence against the possibility of Ni(II) reduction by the combination of B₂pin₂/LiOMe system, and suggests that the catalytic cycle is initiated through the formation of a Ni(II)-Bpin species” . If Ni(II)-Bpin has formed, then why does the author not observe a shift in the CV?

Our Response: We thank the reviewer for this comment. Upon verifying the data, a misplacement in the original figure was identified and has now been rectified (blue line). As shown below, when B₂pin₂ and LiOMe were added to the Ni(II) catalyst, a shift in the CV was observed. (blue line vs yellow line; purple line vs green line)

If Ni(II) is not reduced by B₂pin₂/base, how does the subsequent oxidative addition with alkyl bromide happen? Does the author propose Ni(II) to Ni(IV) cycle?

Our Response: We appreciate the reviewer's insightful comments. To investigate the reaction mechanism, we conducted a radical trapping experiment (Fig. 6e) and a radical clock experiment (Fig. 6f). A benzyl radical was detected via formation of a TEMPO-trapped adduct, while ring-opening products **86** and **87** were isolated from the reaction of alkyl bromide derivative **85**, supporting the involvement of an alkyl radical pathway.

Based on these mechanistic studies, we propose that the multicomponent reaction is initiated by migratory insertion of the alkyne into a Ni(II)-Bpin species, forming a *syn*-alkenylnickel(II) intermediate (Fig. 7). This species can undergo reversible isomerization to its *anti*-alkenylnickel(II) counterpart. The ligand plays a critical role

Wuhan University

武汉大学

in modulating the relative reactivity of these two intermediates toward the alkyl radical, thereby enabling the selective formation of the carboboration product and a Ni(I) species. The Ni(I) species subsequently reacts with the alkyl halide, generating the alkyl radical and regenerating the Ni(II) catalyst. The proposed catalytic cycle has been added to the revised manuscript.

Proposed mechanism

(6) The formation of a single regioisomer is anticipated with aryl alkynes; however, what drives the alkyl alkyne to provide a single regioisomer? Did the author observe the other regioisomer? (The yield appears to be moderate.)

Our Response: We appreciate the reviewer for these professional comments. Similar to aryl-substituted alkynes, alkyl-substituted alkynes can also direct the regioselectivity of the reaction through steric differentiation between the alkyl and hydrogen termini (alkyl > H). Additionally, no formation of the alternative regioisomer was observed in these reactions and the mass balance was primarily accounted for by the formation of alkyne trimers.

(7) Kinetic studies could assist in identifying the RDS, which will further support the proposed mechanism and existence of equilibrium.

Our Response: We appreciate the reviewer for these professional comments. As suggested, a Hammett plot experiment was conducted. As shown below, the observed positive Hammett ρ values (+0.80 and +0.69) indicate the accumulation of negative charge at the rate-determining step of the reaction, which is consistent with the conclusions proposed in this work.

Wuhan University

武汉大学

Hammett Analysis

Does the reaction proceed through benzyl radical? Did the author employ a radical trapping agent or radical clock substrate to identify the radical nature of the intermediate?

Our Response: We appreciate the reviewer's valuable suggestion. In response, we conducted a radical trapping experiment (Fig. 6e) and a radical clock experiment (Fig. 6f). As shown below, a benzyl radical was detected through the formation of the TEMPO-trapped adduct **83**, and the ring-opening products **86** and **87** were isolated from the reaction with alkyl bromide derivative **85**. These results support the involvement of an alkyl radical intermediate in the reaction mechanism. The corresponding experimental data have been incorporated into the revised manuscript.

Radical trapping experiment

Radical clock experiment

(8) Earlier, the same author published the use of α -bromo carbonyls in anti-selective carboborylation. What happens if you employ α -bromo carbonyls under syn - and $anti$ -conditions? Can the stereodiversity still be obtained?

Wuhan University

武汉大学

Our Response: We thank the reviewer for this valuable suggestion. As shown in Figs. 3 and 4, we have previously applied α -bromophosphates, propargyl bromide, secondary α -bromoesters, α -bromoamides, and allyl bromide under both syn- and anti-selective conditions, successfully affording a range of stereodiverse products. These results have been included in the original manuscript (see Figs. 3 and 4).

Reviewer 2:

The submitted article, entitled “Catalyst geometric variations for stereodivergent synthesis of polysubstituted alkenes,” demonstrates the synthesis of syn and anti-products using Ni catalysts containing various ligand types, efficiently with L4 for syn and L8 for anti-products. Suginome et al. reported nickel-catalyzed cyclizative trans-carboboration of alkynes (Asian J. Org. Chem. 2013, 2, 968 – 976). More recently, Huang et al. reported regioselective trans-carboboration of internal alkynes via a nickel catalysis system with the aid of the directing group strategy (Chem. Sci., 2024, 15, 2236). The regioselectivity was accurately switched by the nitrogen ligand (terpy) and phosphine ligand (Xantphos). Based on these literatures it appears that the use of Ni catalyst for the reported reaction is not an unusual. Author must include these references. Yin et al. (same author) reported asymmetric anti-selective borylalkylation of terminal alkynes by nickel catalysis with wide substrate scope (Ref. 40). In my opinion, this report is showing some new aspects of Ni-catalyzed regioselective carboboration of alkynes but it lacks novelty and is not suitable for the publication in Nature Communications.

Our Response: While we respect the comment, we would like to clarify the unique

Wuhan University

武 汉 大 学

conceptual and methodological advances of this work, which we believe address critical gaps in the field:

(1) A new paradigm in ligand effect and catalyst design: Rather than focusing on the development of novel ligand scaffolds, our study demonstrates for the first time that subtle ligand modifications can systematically regulate the geometry of the catalytic center, with profound effects on reaction selectivity. By establishing a direct correlation between ligand-induced geometric perturbations and key elementary steps in metal catalysis, we provide a predictive and generalizable framework for understanding and tuning catalytic behavior. This concept—linking fine ligand tuning to mechanistic control—offers a new direction for rational catalyst design, which has been largely underexplored in prior studies.

(2) Advancement in stereoselective control: Our study is centered on stereoselectivity rather than regioselectivity. As noted by Reviewer #1, “*The syn-selective carbofunctionalization, including carboborylation of alkynes, is well known; however, anti-selective carboborylation is uncommon.*” Current strategies for achieving anti-selectivity typically rely on intermolecular interactions and are governed by coordinating groups, steric effects, or radical pathways—approaches that often impose strict limitations on substrate scope. In contrast, our work introduces a fundamentally distinct strategy by controlling stereoselectivity via catalyst. This approach enables broad substrate generality, accommodating aryl, alkyl, and internal alkynes, thereby overcoming a long-standing limitation in alkyne functionalization and expanding synthetic utility.

(3) Synthetic utility: This method provides direct access to synthetically valuable architectures that are otherwise not easily accessible through established methodologies.

Collectively, this work not only represents a significant advancement in alkyne carboboration chemistry but also introduces a transformative strategy for leveraging metal–ligand interactions to achieve stereocontrol in catalysis. We therefore believe that our study meets the high standards of *Nature Communications*.

Specific comments:

(1) In introduction section, authors should include relevant references related to organoboron compounds and polysubstituted vinyl boronates.

Our Response: We thank the reviewer for his/her good suggestion, and these relevant references have been added in the revised manuscript. Please see ref. 41-42.

(2) References related to carboboration reactions must be included, particularly related to Ni-catalyzed.

Our Response: We thank the reviewer for his/her good suggestion, and these references on relevant carboboration reactions have been added in the revised

Wuhan University

武汉大学

manuscript. Please see ref. 38-48.

(3) There are no phosphorous based ligands explored towards this alkyne carboboration. Any reason why nitrogen ligand over phosphorous.

Our Response: We thank the reviewer for this valuable suggestion. In response, we systematically evaluated the performance of monophosphine (**L25**, **L26**), diphosphine (**L27**, **L28**), and aminophosphine ligands (**L29**, **L30**). The results indicate that phosphine-based ligands are not suitable for this alkyne carboboration reaction, instead, they are more likely to induce alkyne trimerization. Instead, they exhibit a pronounced tendency to induce alkyne trimerization. This observation is likely attributable to the stronger electron-donating capacity of phosphine ligands relative to their nitrogen-based counterparts, which facilitates cyclometalation and thereby promotes trimer formation. These findings have been incorporated into the revised Supplementary Information.

(4) Most of the carboboration product show moderate yields, is there any byproducts arising from the competent 2 component system or is it just starting material,

Our Response: We thank the reviewer for this comment. In response, we carefully analyzed the reactions that afforded low yields. As shown below, alkyne trimers were identified as the major side products in two representative cases. Additionally, we observed protodehalogenation products and homocoupling of electrophiles. Notably, no byproducts derived from the corresponding two-component systems were detected in these reactions. A detailed discussion has been added to the main text, and the

Wuhan University

武汉大学

corresponding experimental results have been included in the revised Supplementary Information (Section 5, page S26).

if so higher temperature or time or catalyst loading could help?

Our Response: We thank the reviewer for this valuable suggestion. In response, we attempted to optimize the reactions that afforded moderate yields. However, as shown below, variations in temperature, catalyst loading, and reaction time did not lead to any significant improvement in yield.

Entry	Variants	70 Yield (%)	anti:syn
1	none	46	>20:1
2	50 °C instead of 40 °C	44	>20:1
3	60 °C instead of 40 °C	40	>20:1
4	30 °C instead of 40 °C	25	>20:1
5	24 h instead of 12 h	46	>20:1
6	Ni/ L5 (10 mol%) instead of Ni/ L5 (5 mol%)	42	>20:1
7	Ni/ L5 (2.5 mol%) instead of Ni/ L5 (5 mol%)	30	>20:1

(5) Substrate scope of electrophile is very limited (only activated benzyl bromide), authors should explore unactivated alkyl halide for carboboration reaction. MeI should explore?

Wuhan University

武 汉 大 学

Our Response: We thank the reviewer for this comment. The electrophile scope in our study is not limited to activated benzyl bromides; it also includes α -bromophosphates, propargyl bromide, secondary α -bromoesters, α -bromoamides, and allyl bromide. These results were presented in the original manuscript (see Figs. 3 and 4).

Additionally, we evaluated less reactive electrophiles such as phenylpropyl halide and methyl iodide under the standard conditions. However, the desired three-component coupling products were not observed. Instead, alkyne trimers and dehalogenation byproducts were identified as the major reaction outcomes.

(6) Authors should provide NMR evidence for Ni(II)-Bpin species, after the addition of B_2pin_2 and LiOMe to the Ni catalyst.

Our Response: We thank the reviewer for this comment. We tried to collect the NMR data of Ni(II)-Bpin species but failed due to its instability.

(7) Cyclic voltammogram that author presented is questionable. They claimed that after the addition of B_2pin_2 and LiOMe to Ni catalyst, it is closely matched with the reported literature, but with the ligand it must have some change that can be seen with some difference. Authors can comment on this?

Our Response: We thank the reviewer for this valuable comment. Upon ligand addition, the signal peaks do exhibit a shift relative to literature reports, arising from the ligand altering the electronic density and geometric structure of the nickel center via electronic and steric effects—thereby modifying its redox potential. However, the characteristic trend of the peaks is consistent with that of divalent nickel species.

Difference is more when B_2pin_2 and LiOMe were used in excess (4 equiv.). It is clear in CV, both peaks around -1.1 and -1.7 V are due to ligand effect. It needs to justify?

Our Response: We thank the reviewer for this comment. We have added a cyclic voltammetry (CV) curve for $NiCl_2 \cdot DME$ with B_2pin_2 and LiOMe (red line). As shown below, the peak position shifts significantly in the absence of ligand compared to its

Wuhan University

武 汉 大 学

position in the presence of ligand, confirming that this shift is attributable to the ligand effect.

Additionally, if CV data collected on 200 mV sec⁻¹ scan rate, then also current reaches up to 0.2 μ A?

Our Response: We thank the reviewer for pointing out this error. The correct unit of current is ampere (A), not milliampere (mA), and this has been corrected in the revised manuscript.

(8) Some of experimental data should be recheck, ¹³C NMR of 4, spectrum frequency is not according to 600 MHz instrument? Are there compound 26 data collected on 400 MHz? For compound 29, if it is on 400 MHz, then ¹¹B cannot be 193 MHz? Similarly, data need to be checked for more clarity.

Our Response: We appreciate the reviewer for this comment. Our experimental data were collected using JNM-ECZ 400 and Bruker 600 MHz instruments. We have rechecked the experimental data and made corrections.

(9) HRMS data missing for a few compounds like for 6, 19, 24.

Our Response: We thank the reviewer for this comment. The HRMS data for compound 6, 19, 24 have been added in the revised SI.

(10) In DFT, author work on optimized gas phase structures along with solvent effect. Can author comment on this, why the gas phase condition considers even all reactions performed at 40°C in 1,4-dioxane? Without solvent, what changes appear in the product as per DFT?

Our Response: We thank the reviewer for this professional comment. Typically, the

Wuhan University

武汉大学

shape of neutral system's potential surface is similar in both gas-phase and solution-phase. This was confirmed by optimizing key nickel-alkenyl intermediates ($^{\text{S}}\text{CP3}$, $^{\text{T}}\text{CP3}$, $^{\text{S}}\text{CP4}$, $^{\text{T}}\text{CP4}$, $^{\text{S}}\text{CP7}$, $^{\text{T}}\text{CP7}$, $^{\text{S}}\text{CP8}$, $^{\text{T}}\text{CP8}$) under the SMD model with 1,4-dioxane as the solvent. The root-mean-square deviation between the gas-phase and solution-phase optimized structures is shown in the following figure, which demonstrates high degree of overlap. Therefore, it's reasonable to optimize structures in gas-phase and calculate highly accurate single-point energies with solvation correction to get a good balance between computational cost and calculation accuracy.

CP	RMSD (Å)	$G_{\text{sol-opt}}^{\text{a}} - G_{\text{gas-opt}}^{\text{b}}$ (kcal/mol)
$^{\text{T}}\text{CP3}$	0.178	0.73
$^{\text{T}}\text{CP4}$	0.060	-0.65
$^{\text{S}}\text{CP3}$	0.062	0.13
$^{\text{S}}\text{CP4}$	0.322	-0.49
$^{\text{S}}\text{CP7}$	0.082	-0.07
$^{\text{S}}\text{CP8}$	0.042	1.05
$^{\text{T}}\text{CP7}$	0.443	0.43
$^{\text{T}}\text{CP8}$	0.049	-0.03

^aThe energy was calculated by M06/def2-TZVP/SMD(1,4-dioxane)// B3LYP-D3(BJ)/def2-SVP/SMD(1,4-dioxane)

^bThe energy was calculated by M06/def2-TZVP/SMD(1,4-dioxane)// B3LYP-D3(BJ)/def2-SVP

(11) The DFT provides better insights on how the geometry influences the selectivity of the target product. However, the work lacks in proving better insights on the mechanism via experimental route. (refer <https://doi.org/10.1016/j.chempr.2022.10.003>) Also, no explanation or insights are

Wuhan University

武 汉 大 学

provided on why the optimized ligand provides selectively one single isomer.

Our Response: We do thank the reviewer for this comment. While we have not yet characterized the product intermediate via single-crystal structural analysis, we have verified the feasibility of this mechanism through a series of mechanistic experiments, including isomerization experiments, quenching experiments, stoichiometric experiments, radical trapping experiments, and radical clock experiments. Based on the above experimental results, we validated the rationality of this mechanism through DFT calculations, providing a comprehensive and in-depth understanding of the reaction mechanism.

As noted in our original manuscript, the stereoselectivity of the reaction is primarily determined by the steric hindrance effect of the ligand (Fig. 8). In the catalytic system with **L4** (Fig. 8c), the steric hindrance from the 6-substituent of the ligand allows the divalent alkenylnickel complex to adopt either a tetrahedral or a distorted square pyramidal coordination geometry with a triplet (high-spin) state. Both configurations create a crowded environment, making the radical addition step sensitive to the stereochemistry of the alkenyl ligand; consequently, the syn-configuration is favored due to its lower overall steric hindrance, resulting in the observed syn-selectivity. In contrast, in the reaction system with **L8**, the alkenylnickel complex preferentially adopts a square-planar coordination geometry with a singlet (low-spin) state (Fig. 8d). Unlike the tetrahedral and distorted trigonal bipyramidal coordination geometries, the square-planar geometry results in a less congested environment. As a result, the radical addition step is minimally influenced by the stereochemistry of the alkenyl ligand, and stereoselectivity is dominated by the reductive elimination step.

Reviewer 3:

The authors describe a new method for diastereodivergent synthesis of tri- and some tetrasubstituted alkenes through nickel-catalyzed boroalkylation of terminal (and some internal) alkynes. In the manuscript, the authors describe the development of the reaction, exploration of the substrate scope, synthetic applications, and an experimental and theoretical study of the reaction mechanism. The main contribution described in the manuscript is the use of selective isomerization of the alkenyl metal intermediate to achieve diastereodivergent boroalkylation. This is a significant advance and as a result the manuscript merits publication in Nature Communications. The work described in the manuscript is comprehensive and well executed. The substrate scope is well explored and shows some of the downsides of the approach. The selectivities are moderate (albeit still useful) with a range of terminal alkynes. On the other hand, the scope of tetrasubstituted alkynes runs into known limitations of the carboboration chemistry in controlling the regioselectivity. Finally, the alkylation partners exhibit limitations common for this type of chemistry. Synthetic applications

Wuhan University

武 汉 大 学

are meaningful, and the mechanistic studies are informative. I am not able to fully evaluate the DFT part of the mechanistic study and leave that to other reviewers.

Before publication the authors should address the following issues:

Our response: We thank the reviewer for his/her positive comments and support for publishing our work in *Nature Communications*.

(1) The introduction is very broad, and it is easy to lose sight that this reaction is a modification of a well-established carboboration or carboalkylation of alkynes. A part of the introduction should provide a short summary of that specific field. For example, some key references that provide a useful benchmark are the following references: J. Am. Chem. Soc. 134, 15165-15168 (2012), J. Am. Chem. Soc. 138, 7528-7531 (2016), ChemCatChem 7, 2108-2112 (2015), and a variety of reviews such as: Eur. J. Org. Chem. 2022, e202200521 and other reviews.

Our response: We appreciate the reviewer's professional suggestions. We have added a summary of alkyne carboboration field in the revised manuscript. The mentioned key references have been included in the revised manuscript.

(2) The title is confusing, and the authors should change it. Suggestion: Catalyst controlled stereodivergent synthesis of polysubstituted alkenes.

Our response: We appreciate the reviewer for this suggestion and have accordingly revised the title.

(3) In line 39, the authors state: "However, the effective use of these properties as design elements for reaction development remains underexplored". One can argue that any successfully developed catalytic reaction is an example of it. Also, I am not sure what would make something a "design element" vs a "reaction variable" that is optimized together with other reaction variables. The authors should eliminate or change this sentence.

Our response: We appreciate the reviewer for this suggestion and have revised this sentence.

Reviewer 4:

Yin and co-workers reported an interesting ligand-controlled stereodivergent borylalkylation of alkynes. The syn- and anti-carboboration products were obtained by using the sterically hindered 2,2'-bipyridine ligand (**L4**) and the iPr-Pmrox type ligand (**L8**), respectively. Through experimental and computational analysis, they claimed that the divergent stereoselectivities are attributed to the different geometric configurations of Ni catalysts. However, their presented computational results do not support their conclusions and the explanations remain vague.

Wuhan University

武 汉 大 学

Our response: We thank this reviewer for the above positive comments and the following mechanistic suggestions, which greatly help us to improve the quality of this manuscript. Accordingly, we conducted a comprehensive computational analysis, which are all consistent with our previous proposed mechanism.

Main concerns are listed below.

(1) Their explanation for the stereoselectivity is based on the Curtin-Hammett principle. However, the activation barrier of *syn*/*anti* isomerization is much higher than that of the subsequent steps, which indicates the reaction does not fulfill the Curtin-Hammett principle.

Our response: We appreciate the reviewer's valuable comments and apologize for any confusion caused. In our calculations, we employed the widely accepted E/Z isomerization mechanism involving a three-membered η^2 -coordinated transition state. The computed energy barriers are consistent with those reported in previous studies (Fig. 8a). Notably, the generation of the benzyl radical in our system involves a significant energy barrier, resulting in a low steady-state concentration of radical species under the reaction conditions. This barrier is not fully reflected in the subsequent reaction potential energy surface. (Figure 8b) Therefore, we propose that, prior to oxidative addition of the benzyl radical, the divalent nickel species can freely interconvert between *syn*- and *anti*-configurations. The stereoselectivity is thus primarily governed by the subsequent steps, a conclusion supported by our mechanistic experiments (Fig. 6). We have revised the manuscript for better clarification.

(2) The geometric configurations of Ni catalysts were considered to be crucial for the reaction. Have the authors systematically analyzed all possible geometric configurations of Ni complexes, including the intermediates and transition states? For instance, interchanging the positions of the Cl, Bn, and alkenyl groups in CP5 will lead to different geometric configurations.

Our response: We appreciate the reviewer's valuable comments. With the aid of the XTB-6.5.1 and Molclus-1.12 program, we have systematically investigated the conformations of key nickel complexes, including intermediates and transition states. We have selected some representative results, which are presented in the following figure.

Wuhan University

武汉大学

(3) Related to the previous question, the reviewer noticed that CP10 adopted a different geometric configuration compared to CP9. CP10 has to undergo configuration isomeriation for the reductive elimination step; however, CP9 can directly undergo reductive elimination. Why did the authors use different configurations here?

Our response: We thank the reviewer for the professional comments. As shown in the figure above (Please refer to the figure in response to Question 2.), we have

Wuhan University

武汉大学

evaluated multiple possible configurations of the nickel complex and confirmed that the currently proposed pathway is the most reasonable, both energetically and structurally. Computational results indicate that **CP9** can undergo direct reductive elimination, whereas a direct C–C bond-forming transition state from **CP10** could not be located during optimization. To address this, starting from **CP10**, we first performed a flexible scan of the C–C bond formation process using the direction of C–C bond formation as the reaction coordinate. Subsequently, the maxima identified in the flexible scan was further optimized, leading to **TS10**. This process actually involves two steps: **CP10** first undergoes a conformational change to form **CP10-B**, followed by reductive elimination.

Will the configuration affect the activation barrier?

Our response: We appreciate for this insightful comment from the reviewer and fully endorse it. Intermediates in different configurations exhibit substantial differences in their reaction barriers. Notably, although **TS10-C** is identified as the lowest-energy reductive elimination transition state, key intermediate **CP10** is unable to convert to **CP10-C**, the precursor of **TS10-C**, according to molecular dynamics studies. This is likely due to the coordination between the oxygen atom of the Bpin moiety and the nickel center, which restricts the free configurational interconversion of the trivalent nickel intermediate. As a result, the system is confined to a limited subset of accessible structures, such as the transformation from **CP10** to **CP10-B**.

The authors should carefully consider all possible configurations, otherwise the comparison is biased and the conclusion that reductive elimination step decides the stereoselectivity in the anti-selective reaction is not reliable.

Our response: We thank the reviewer for the professional comments. We have

Wuhan University

武 汉 大 学

conducted a comprehensive investigation into all possible configurations (Please refer to the figure in response to Question 2). On the basis of such investigation, we carefully drew the conclusion.

(4) For the same reason, the distortion–interaction analysis (Figure S13) of the reductive elimination is not convincing. It’s unsurprising that TS10 experiences larger distortion than TS9, since TS10 undergoes configuration isomerization compared to CP10 as I pointed out above.

Our response: We appreciate the reviewer’s professional comments. We agree that performing a distortion–interaction energy analysis on two reductive elimination transition states with markedly different structures is not appropriate. Accordingly, this section has been removed from the revised manuscript and Supplementary Information.

(5) Why do the two ligands, **L4** and **L8** lead to different preferred geometric configurations and preferred spin states for the Ni catalysts?

Our response: According to crystal field theory, the electronic configurations of transition metal complexes are highly sensitive to their geometric structures. The 16-electron nickel complexes investigated in this study primarily exist in either singlet or triplet states. Singlet nickel complexes—commonly found in nickel–carbon systems—feature all paired electrons, with four fully occupied d-orbitals and one empty d-orbital, typically adopting a square planar geometry. In this arrangement, the highest-energy antibonding orbital remains unoccupied, contributing to overall stabilization. We propose that such a square planar geometry is present in the **L8** complex.

In contrast, the **L4** complex experiences significant steric repulsion due to the proximity of a methyl substituent, which destabilizes the square planar geometry. A shift to a high-spin tetrahedral configuration can alleviate this steric strain, making the triplet state more favorable. These structural and electronic preferences are consistent with previous reports (see refs. 78–79: *J. Am. Chem. Soc.* 146, 3043–3051 (2024); *ACS Catal.* 14, 6897–6914 (2024)).

(6) The explanation for the 1.6 kcal/mol difference between TS3 and TS4 stays vague. The statement “the distorted trigonal bipyramidal geometry, which exhibits greater stretching than the tetrahedral geometry, is more prone to be attacked by free radicals” needs more support, such as structural analysis.

Our response: We appreciate the reviewer’s professional comments. We apologize for any misunderstanding caused by our imprecise wording. What we intended to convey is that the 1.6 kcal/mol energy barrier difference between TS3 and TS4 reflects the different abilities of **CP3** and **CP4** to be attacked by benzyl radical. Through distortion/interaction analysis, we found this difference comes from their

Wuhan University

武 汉 大 学

distinct steric hindrance distributions, which was further corroborated by the surface distance projection map. (Fig S14) And we never intended to make a categorical inference that “distorted trigonal bipyramidal geometry is more prone to be attacked by free radicals than tetrahedral geometry” We have revised the manuscript for better clarification.

(7) The ligand screening experiments in Figure 2 demonstrate that the substituents also play an important role in controlling the stereoselectivity, for instance, reversed stereoselectivities were obtained by using L4 and L5 lead and using L6 and L7, respectively. Can these substituent effects be explained by their mechanistic model?

Our response: We thank this reviewer for this comment. The current mechanistic model accurately accounts for the stereodiversity effects of ligand substituents. Specifically, *anti*-stereoselectivity is observed with ligands **L5** and **L7** (which lack ortho steric substituents), whereas *syn*-stereoselectivity is seen with ligands **L4** and **L6** (bearing ortho methyl groups). This trend aligns well with our proposed mechanism:

- For ligands without ortho steric substituents (**L5** and **L7**), the key Ni(II) intermediate tends to adopt a low-spin square geometry. In such cases, the *syn* and *anti*-pathways of the oxidative addition step exhibit negligible energy barrier differences, and stereoselectivity is dominated by the reductive elimination step, which is leading to an *anti*-selectivity—consistent with the observed results.
- For ligands with ortho steric substituents (**L4** and **L6**), the key Ni(II) intermediate prefers a high-spin tetrahedral configuration. Here, stereoselectivity is determined by the oxidative addition step: since radical binding to nickel is sterically sensitive, the *syn*- configuration is favored due to its lower overall steric hindrance, resulting in the observed *syn*-selectivity.

Other comments:

(1) The legend of Fig.1c (Catalyst geometric variations for stereodivergent borylalkylation of alkynes) seems inappropriate, as it's hard to see the difference in catalyst geometries for the two systems.

Our response: We appreciate the reviewer's professional comments. We have revised the legend of Fig.1c.

(2) Page 4, line 82, it should be “ligand (L4)” not “ligands (L4)”

Our response: We appreciate the reviewer's professional comments. We have corrected this error in the revised manuscript.

(3) L5 and L6 are not mentioned in the main text.

Wuhan University

武 汉 大 学

Our response: We appreciate the reviewer's professional comments. We have mentioned these ligands in the revised manuscript.

(4) The geometric configurations in Fig 7 are unclear and difficult to differentiate, especially for TCP4 and TCP8. The authors label TCP4 as trigonal bipyramidal and TCP8 as square pyramidal, but they appear identical in drawing. The authors should carefully revise the structures in Fig 7 to ensure that the ligands in the apical/axial and equatorial positions are clearly distinguishable.

Our response: We appreciate the reviewer's professional comments. We have revised the structures for greater clarity. We hope these changes meet your expectations.

Moreover, based on the coordinates in the SI, the geometric configuration of TCP4 is closer to square pyramidal. The deviation of Cl from the equatorial plane appears to be due to steric hindrance at the Bpin position.

Our response: We appreciate the reviewer's professional comments. We have re-examined the structure of TCP4 and revised its label from "trigonal bipyramidal" to "square pyramidal".

(5) What is the spin state for TCP8? The label for spin state is missing in TCP8.

Our response: We thank the reviewer for this comment. We have corrected this error in the revised manuscript.

(6) The writing may need further improvement. Some sentences contain grammatical errors or are unclear. For example, the sentence "Although it relatively high energy, the radical chain initiation step surmounts a comparable energy barrier (Fig. 7b), and the free radicals need to cross solvent cages, leading to a low concentration of alkyl radicals." contains a grammatical error and should be revised for clarity. Additionally, in the sentence "We also discovered the reactivity of Ni(II) complex with different configurations which are raised by different spin states," the word "discovered" may be better replaced with "studied" or "investigated".

Our response: We thank the reviewer for pointing out these errors. We have revised the manuscript accordingly and carefully reviewed the entire text to ensure accuracy and clarity.

Response to Reviewers' Comments

Reviewer #1 (Remarks to the Author):

Authors addressed the concerns; consequently, I recommend the manuscript for publication in its current state.

Our Response: We deeply appreciate the reviewer for his/her constructive suggestions and positive comments to improved enhancement of our research. We are very pleased that our reply has satisfied the reviewers.

Reviewer #2 (Remarks to the Author):

The submitted revised article, entitled “Catalyst geometric variations for stereodivergent synthesis of polysubstituted alkenes,” demonstrates the synthesis of syn and anti-products using Ni catalysts containing various ligand types, efficiently with L4 for syn and L8 for anti-products. The authors have addressed most of the technical concerns raised by the reviewers; however, my primary concern pertains to the novelty of the work.

Our Response: Thanks for your comment. The novelty of our work lies in revealing that subtle ligand modifications can systematically regulate the geometric structure of catalytic centers, thereby exerting a profound impact on reaction selectivity and enabling precise control over the stereoconfiguration of alkenes, which has been largely underexplored in prior studies. Notably, the novelty, significance, and broad relevance of this work to the readership have been explicitly affirmed by the other three reviewers. Their positive assessments highlight the potential contributions and value of our research to the field and the readership.

Reviewer #4 (Remarks to the Author):

The authors have addressed my questions. I support its publication on Nature Communications.

Our Response: We sincerely thank you for your thorough review of our revised manuscript and for your positive feedback. Your valuable comments were crucial in helping us improve our work, and we greatly appreciate your recommendation to publish our research in Nature Communications.

General Discussion

The paper discusses the nickel-catalyzed stereodivergent carboborylation of alkynes, demonstrating the possible role of the catalyst's geometry in modulating reaction stereoselectivity. The synthetic utility of this chemistry is demonstrated by the synthesis of drug-relevant molecules. A detailed mechanistic study including DFT calculations was carried out to support the proposed mechanism. Overall, the ligand-controlled stereodivergent synthesis is impressive.

Recommendation

The syn-selective carbofunctionalization including the carboborylation of alkynes, is well known, however, the anti-selective carboborylation is uncommon. The same author reported the anti-selective carboborylation of alkynes (10.1021/jacs.3c05969). The difference is a change in the electrophile; i.e., α -bromo carbonyls vs. benzyl bromides. Since the current work combines strategies and demonstrates the ligand-controlled stereodiversity, also provides a detailed mechanistic study, I recommend it for publication after a major review.

Comments

In reaction development, the author should disclose the byproducts observed during the optimization. In Fig. 3 and Fig. 4, the reason for lower yields for substrates like **70** was not discussed. Did the author observe byproducts?

Apart from activated halides, did the author attempt a reaction with unactivated alkyl halides (bromides or iodides)?

Page number 10: line 172-174, the author stated that "*These results strongly suggest the involvement of a rapid and reversible isomerization process in the catalytic cycle, which likely occurs prior to the oxidative addition step, and are consistent with the Curtin–Hammett principle*". This is a bit confusing for me because the isomerization occurs from the vinylnickel intermediate; even if it reacts with water or benzylbromide, the stereochemical outcome should be the same because the ligand is what dictates the stereochemistry (See Fig.6c and 6d).

Page number 10: line 178, the author states that "*This finding provides evidence against the possibility of Ni(II) reduction by the combination of B₂pin₂/LiOMe system, and suggests that the catalytic cycle is initiated through the formation of a Ni(II)-Bpin species*". If Ni(II)-Bpin has formed, then why does the author not observe a shift in the CV? If Ni(II) is not reduced by B₂pin₂/base, how does the subsequent oxidative addition with alkyl bromide happen? Does the author propose Ni(II) to Ni(IV) cycle?

The formation of a single regioisomer is anticipated with aryl alkynes; however, what drives the alkyl alkyne to provide a single regioisomer? Did the author observe the other regioisomer? (The yield appears to be moderate.)

Kinetic studies could assist in identifying the RDS, which will further support the proposed mechanism and existence of equilibrium.

Does the reaction proceed through benzyl radical? Did the author employ a radical trapping agent or radical clock substrate to identify the radical nature of the intermediate?

Earlier, the same author published the use of α -bromo carbonyls in anti-selective carbonylation. What happens if you employ α -bromo carbonyls under syn- and anti-conditions? Can the stereodiversity still be obtained?

Supplementary Information

General comment: The SI looks good and clean.

The authors describe a new method for diastereodivergent synthesis of tri- and some tetrasubstituted alkenes through nickel-catalyzed boroalkylation of terminal (and some internal) alkynes. In the manuscript, the authors describe the development of the reaction, exploration of the substrate scope, synthetic applications, and an experimental and theoretical study of the reaction mechanism.

The main contribution described in the manuscript is the use of selective isomerization of the alkenyl metal intermediate to achieve diastereodivergent boroalkylation. This is a significant advance and as a result the manuscript merits publication in *Nature Communications*.

The work described in the manuscript is comprehensive and well executed. The substrate scope is well explored and shows some of the downsides of the approach. The selectivities are moderate (albeit still useful) with a range of terminal alkynes. On the other hand, the scope of tetrasubstituted alkynes runs into known limitations of the carboboration chemistry in controlling the regioselectivity. Finally, the alkylation partners exhibit limitations common for this type of chemistry.

Synthetic applications are meaningful, and the mechanistic studies are informative. I am not able to fully evaluate the DFT part of the mechanistic study and leave that to other reviewers.

Before publication the authors should address the following issues:

1. The introduction is very broad, and it is easy to lose sight that this reaction is a modification of a well-established carboboration or carboalkylation of alkynes. A part of the introduction should provide a short summary of that specific field. For example, some key references that provide a useful benchmark are the following references: *J. Am. Chem. Soc.* 134, 15165-15168 (2012), *J. Am. Chem. Soc.* 138, 7528-7531 (2016), *ChemCatChem* 7, 2108-2112 (2015), and a variety of reviews such as: *Eur. J. Org. Chem.* **2022**, e202200521 and other reviews.
2. The title is confusing, and the authors should change it. Suggestion: Catalyst-controlled stereodivergent synthesis of polysubstituted alkenes.
3. In line 39, the authors state: "However, the effective use of these properties as design elements for reaction development remains underexplored". One can argue that any successfully developed catalytic reaction is an example of it. Also, I am not sure what would make something a "design element" vs a "reaction variable" that is optimized together with other reaction variables. The authors should eliminate or change this sentence.